# Changing European Hydroclimate under a Collapsed AMOC in the Community Earth System Model

René M. van Westen<sup>1</sup>, Karin van der Wiel<sup>1</sup>, Swinda K.J. Falkena<sup>2</sup>, and Frank Selten<sup>1</sup>

Correspondence: René M. van Westen < r.m. van westen @uu.nl>

Abstract. The Atlantic Meridional Overturning Circulation (AMOC) is expected to weaken or even collapse under anthropogenic climate change. Given the importance of the AMOC in the present-day climate, this would potentially lead to substantial changes in the future projections of the impacts of climate change on regional weather, which is highly relevant for society. Precipitation rates over Europe are expected to decrease under an AMOC collapse, potentially affecting the European hydroclimate. Here, we analyse the impacts of different AMOC collapse and climate change scenarios on the European hydroclimate in a unique set of AMOC experiments executed with the fully-coupled Community Earth System Model (CESM). In general, drier hydroclimatic conditions are expected under an AMOC collapse. The dominant drivers of this change depend on the specific combination of AMOC strength and radiative forcing. In AMOC collapse scenarios under pre-industrial conditions the dominant driver are reduced precipitation rates over the entire European continent. AMOC collapse in combination with increased radiative forcing (RCP4.5, RCP8.5) also leads to higher potential evapotranspiration rates, which further exacerbates the noted shifts to increased seasonal drought (extremes). Here, AMOC collapse enhances well-documented shifts to a drier summer climate in Europe in 'standard' projections of future climate change. In summary, these results indicate a considerable influence of the AMOC on future European hydroclimate. It is therefore vital that climate change projections of European hydroclimate for the (far) future consider the possibility of AMOC changes, and the exacerbated effects this would have on projected regional hydrological changes and consequences for ecosystems and society.

#### 1 Introduction

The Atlantic Meridional Overturning Circulation (AMOC) is a key focus of current climate research due to its crucial role in regulating the global climate (Srokosz and Bryden, 2015). The present-day AMOC carries about 1.5 PW of energy (at 26°N) northward, which effectively cools the Southern Hemisphere and warms the Northern Hemisphere (Johns et al., 2011). The AMOC is considered a potential climate tipping element, meaning that it can undergo a transition from a relatively strong overturning state to a much weaker one (Armstrong McKay et al., 2022). The northward heat transport reduces 75 % in a scenario where the AMOC completely collapses and this altered heat transport induces widespread changes in regional and global climate patterns (Orihuela-Pinto et al., 2022; Bellomo et al., 2023; van Westen et al., 2024b).

<sup>&</sup>lt;sup>1</sup>Royal Netherlands Meteorological Institute (KNMI), De Bilt, the Netherlands

<sup>&</sup>lt;sup>2</sup>Institute for Marine and Atmospheric research Utrecht, Department of Physics, Utrecht University, Utrecht, the Netherlands

Previous studies have analysed the climate responses under substantially weaker AMOC strengths in fully-coupled global climate models (GCMs). This weaker AMOC state is achieved by applying a (large) freshwater flux forcing over the North Atlantic Ocean for several decades (Jackson et al., 2023), after which the climate responses are compared to a reference simulation without a freshwater flux forcing. On a planetary scale, the Northern Hemisphere cools while the Southern Hemisphere warms, the tropical rain bands migrate southward, and there is a redistribution of the dynamic sea level (Vellinga and Wood, 2002; Levermann et al., 2005; Jackson et al., 2015; Orihuela-Pinto et al., 2022; Bellomo et al., 2023; van Westen et al., 2025a). There are a few regions where AMOC fluctuations induce striking climate responses. For example, in the northern Amazon Rainforest, an AMOC collapse leads to a delayed seasonal cycle and an intensification of the dry season (Ben-Yami et al., 2024). There is also evidence that changes in the AMOC can induce far-field hydroclimate responses over the Australasian region, for example the southeastern portion (New Zealand) is experiencing drier conditions throughout the year (Saini et al., 2025). The European region shows relatively large climate responses under a collapsing AMOC, as the climate strongly cools as a consequence of the reduced meridional heat transport and expanding Arctic sea-ice pack (van Westen et al., 2024b).

Beyond this mean cooling of European climate under an AMOC collapse, Europe is likely to experience more intense cold extremes and winter storms, stronger westerlies, and reduced precipitation rates (Jacob et al., 2005; Brayshaw et al., 2009; Jackson et al., 2015; Bellomo et al., 2023; Meccia et al., 2024; van Westen and Baatsen, 2025). To our knowledge, there have been no studies that analyse the effects of AMOC collapse on the European (summer) hydroclimate, including the occurrence of droughts. Such quantifications of the changes to the future European hydroclimate are crucial however, as societies and ecosystems depend on water in many ways (Lee et al., 2025). Global warming is projected to cause changes in mean seasonal precipitation and evaporation rates (Cook et al., 2020), as well as cause intensification of (multi-year) droughts (van der Wiel et al., 2023) and floods. Given that an AMOC collapse scenario leads to further reductions of seasonal mean precipitation beyond 'standard' projections of climate change (Bellomo et al., 2023), we hypothesise that projected changes in hydrological extremes are also exacerbated under an AMOC collapse. In line with this hypothesis, Ionita et al. (2022) demonstrated intensification of droughts over Europe under AMOC weakening over the historical period. The drought intensification decreases agricultural land production from boreal spring to summer, as was specifically shown for the United Kingdom under an AMOC collapse (Ritchie et al., 2020) or Subpolar Gyre collapse (Laybourn et al., 2024).

The goal of this paper is to provide a quantitative picture of how the balance between precipitation and potential evapotranspiration changes under different AMOC regimes. We will use a set of simulations using the fully-coupled Community Earth System Model (CESM, version 1.0.5) that resulted in eight different scenarios for the AMOC in combination with (future) radiative forcing (van Westen et al., 2024a, 2025b). These include collapsed AMOC regimes under pre-industrial conditions, and under the Representative Concentration Pathways (RCP) 4.5 and 8.5 scenarios. An extensive analysis of the effect of an AMOC collapse on the European temperature extremes was already conducted for this unique set of CESM simulations (van Westen and Baatsen, 2025), but the effects on the hydroclimate remain unclear. In Section 2, we provide a brief overview of the CESM simulations and how these were analysed. Next in Section 3, we will analyse the changes in the European hydroclimate and provide the physical mechanisms behind these changes. The results are discussed and summarised in the final Section 4.

## 2 Methods

#### 2.1 The CESM Simulations

The CESM version analysed here has horizontal resolutions of 1° for the ocean/sea ice and 2° for the atmosphere/land components, respectively. We applied a freshwater flux forcing (F<sub>H</sub>) over the latitude bands between 20°N to 50°N in the Atlantic Ocean, which was compensated elsewhere (at the surface) to conserve salinity. We study eight different scenarios for which the CESM simulations were obtained under constant F<sub>H</sub> and radiative forcing conditions (PI, RCP4.5 and RCP8.5). The details of the simulations are given in Table 1 and more details on how we obtained these simulations are provided below. The simulation name consists of three parts: initially a reference to the radiative forcing conditions, in subscript the applied F<sub>H</sub> strength (in units of ×10<sup>-2</sup> Sv), and in superscript whether the AMOC is in its strong northward overturning state (i.e. 'on') or in a substantially weaker state (i.e., 'off'). When subscripts and/or superscripts are not specified, we refer to the two related simulations (e.g., PI<sup>off</sup> → PI<sup>off</sup><sub>18</sub> and PI<sup>off</sup><sub>45</sub>).

**Table 1.** Overview of the eight different AMOC scenarios simulated with the CESM, including their radiative forcing conditions, freshwater flux forcing strength, AMOC status, time-mean AMOC strength at  $26^{\circ}$ N and 1,000 m depth (expressed in Sverdrups, 1 Sv  $\equiv 10^{6}$  m<sup>3</sup> s<sup>-1</sup>), and time-mean global mean surface temperature (change). The time means are determined over 100-year periods.

| Simulation name                    | Radiative forcing conditions | $F_H$ | AMOC status | AMOC strength | Temperature |
|------------------------------------|------------------------------|-------|-------------|---------------|-------------|
|                                    |                              | (Sv)  |             | (Sv)          | (°C)        |
| PI <sup>on</sup> <sub>18</sub>     | Pre-industrial               | 0.18  | On          | 15.6          | 13.8        |
| $	ext{PI}_{18}^{	ext{off}}$        | Pre-industrial               | 0.18  | Off         | 4.4 (-72%)    | 13.2 (-0.6) |
| RCP4.5 <sup>on</sup> <sub>18</sub> | RCP4.5 at 2100 levels        | 0.18  | On          | 15.5 (-0.6%)  | 16.7 (+2.9) |
| RCP8.5 <sub>18</sub>               | RCP8.5 at 2100 levels        | 0.18  | Off         | 3.3 (-79%)    | 19.2 (+5.4) |
| PI <sup>on</sup> <sub>45</sub>     | Pre-industrial               | 0.45  | On          | 12.4          | 13.7        |
| $	ext{PI}_{45}^{	ext{off}}$        | Pre-industrial               | 0.45  | Off         | 0.1 (-99%)    | 13.2 (-0.5) |
| RCP4.5 $^{ m off}_{45}$            | RCP4.5 at 2100 levels        | 0.45  | Off         | 1.2 (-90%)    | 15.3 (+1.6) |
| RCP8.5 <sub>45</sub> off           | RCP8.5 at 2100 levels        | 0.45  | Off         | 1.2 (-90%)    | 18.8 (+5.1) |

The eight different CESM simulations were obtained as follows. We start from the AMOC hysteresis experiment under constant PI radiative forcing as presented in van Westen and Dijkstra (2023), which is also shown in the inset in Figure 1b. In this experiment, the AMOC was forced under a slowly-varying  $F_H$  at a rate of  $3 \times 10^{-4}$  Sv yr<sup>-1</sup>, which ensures that AMOC changes are primarily caused by intrinsic ocean dynamics. The  $F_H$  was increased up to  $F_H = 0.66$  Sv and the AMOC collapses around  $F_H = 0.525$  Sv. From  $F_H = 0.66$  Sv, the  $F_H$  was then reduced back to zero at the same rate and the AMOC recovers around  $F_H = 0.09$  Sv. This resulted in a multi-stable AMOC regime for 0.09 Sv  $

Figure 1. (a & b): The AMOC strength at 26°N and 1,000 m ocean depth for constant  $F_H = 0.18$  Sv and  $F_H = 0.45$  Sv. Yellow shading indicates the 100-year periods used for the analyses. The inset in panel b shows the quasi-equilibrium hysteresis simulation of van Westen and Dijkstra (2023). (c – j): The yearly-averaged precipitation rates for the different AMOC scenarios. For the PI<sup>off</sup>, RCP4.5 and RCP8.5 scenarios, the yearly-averaged precipitation rates are displayed as the difference compared to their respective PI<sup>on</sup> scenario. Markers indicate non-significant ( $p \ge 0.05$ , two-sided Welch's t-test) differences.

Within the multi-stable AMOC regime, four simulations were branched off under constant  $F_H$  and constant PI radiative forcing conditions. Two simulations were branched from an AMOC on (AMOC off) state at  $F_H = 0.18$  Sv and  $F_H = 0.45$  Sv and were integrated for 500 years. The last 100 years are used for our analyses and are referred to as the PI<sub>18</sub> (PI<sub>18</sub>) and PI<sub>45</sub> (PI<sub>45</sub>), respectively, and are shown in Figures 1a,b. We only consider the last 100 years as they are statistical equilibria of the climate system, which are characterised by time-invariant statistics and any remaining model drift is much smaller than the internal climate variability (van Westen and Baatsen, 2025). These two  $F_H$  values were considered as they demonstrate that statistical equilibria exist close to the AMOC collapse (i.e., PI<sub>45</sub> and PI<sub>45</sub>) and AMOC recovery (i.e., PI<sub>18</sub> and PI<sub>18</sub>), confirming the existence of a broad multi-stable AMOC regime. Note that the AMOC in PI<sub>45</sub> is closer to the tipping point and hence more sensitive under a perturbation than the AMOC in PI<sub>18</sub>.

From the end of  $PI_{18}^{on}$  and  $PI_{45}^{on}$ , van Westen et al. (2025b) branched off the historical forcing (1850 – 1899) followed by either RCP4.5 or RCP8.5 (2006 – 2100) and keeping  $F_H$  fixed. These RCP scenarios were continued beyond 2100 for 400 years to run the AMOC and global climate into a new equilibrium, which was done by fixing their 2100 radiative forcing conditions. The last 100 years are used for our analyses and there is one climate change simulation for which the AMOC recovers, referred to as RCP4.5 $_{18}^{on}$ . The remaining three simulations show an AMOC collapse and are the RCP8.5 $_{18}^{off}$ , RCP4.5 $_{45}^{off}$  and RCP8.5 $_{45}^{off}$ . More details on the AMOC characteristics and responses in these simulations were presented elsewhere (van Westen et al., 2024a, 2025b).

Most results in Section 3 below are presented as follows. The scenarios  $PI_{18}^{on}$  and  $PI_{45}^{on}$  are the reference cases (for their respective  $F_H$  values) and we are interested in the hydroclimate responses for the remaining scenarios. The  $PI_{18}^{off}$ , RCP4.5% and RCP8.5% are presented as differences compared to  $PI_{45}^{on}$ . Similarly, the  $PI_{45}^{off}$ , RCP4.5% and RCP8.5% are compared to  $PI_{45}^{on}$ . For example, the yearly-averaged precipitation rates and responses are shown in Figures 1c – j. These precipitation responses and other hydroclimate responses (see Section 3) are quite similar for  $PI_{45}^{off}$  and  $PI_{45}^{off}$ , and also for RCP8.5% and RCP8.5% and RCP8.5% are robust under  $PI_{45}^{off}$  and RCP8.5% and late of the hydroclimate responses are robust under  $PI_{45}^{off}$  and RCP8.5% atthough the different scenarios ( $PI_{45}^{off}$  vs.  $PI_{45}^{off}$  and RCP8.5% vs. RCP8.5% have slightly different AMOC strengths. The most relevant comparison is made between the RCP4.5% and RCP4.5%, as they have the AMOC in different regimes, which results in opposing precipitation responses (compare Figure 1e and Figure 1i). These two RCP4.5 scenarios will be discussed in greater detail as they represent the hydroclimate under intermediate climate change with AMOC strengths compared to present-day values (RCP4.5%) and under intermediate climate change in combination with a collapsed AMOC (RCP4.5%).

Most variables in these CESM simulations were stored at monthly time intervals, only a limited set of (near-surface) variables were stored at a daily frequency. These include the near-surface (2-meter) air temperature, precipitation rate, and mean sealevel pressure. All other variables analysed in this paper are either analysed at monthly frequency, or statistically downscaled to approximate daily values to enable more detailed analysis of the changing hydroclimate.


## 2.2 Local surface water balance and PET calculations





The local surface water balance (W) is defined as the difference between the local precipitation (P) and local potential evapotranspiration (PET):

$$W(x,y,t) = P(x,y,t) - PET(x,y,t). \tag{1}$$

There are multiple methods to determine the PET, here we follow the procedure outlined in Singer et al. (2021), which uses the Penman-Monteith (Penman, 1948; Monteith, 1965) equation:

PET = 
$$\frac{0.408\Delta(R_n - G) + \gamma\left(\frac{37}{T_a + 273.15}\right)u_2(e_s - e_a)}{\Delta + \gamma(1 + 0.34u_2)},$$
 (2)

where  $R_n$  is the net radiation at the surface, G the soil heat flux,  $\gamma$  the psychrometric constant,  $T_a$  the near-surface (2-meter) air temperature,  $u_2$  the 2-meter wind speed (derived from the 10-meter wind speed, logarithmic profile),  $e_s$  the saturation vapour pressure,  $e_a$  the actual vapour pressure (linked to dew-point temperature,  $T_{\rm dew}$ ), and  $\Delta$  the slope of saturation vapour pressure curve. In Singer et al. (2021), PET was determined using hourly-averaged ERA5 reanalysis data (in units of mm hr $^{-1}$ ) and G was split into a daytime component ( $G^{\rm daytime} = 0.1 \times R_n$ ) and nighttime component ( $G^{\rm nighttime} = 0.5 \times R_n$ ). Note that PET is optimised for well-irrigated grass surface areas and (strongly) overestimates the actual evaporation when soil moisture is depleted. For more details on the PET variables, units and procedure, we refer to Singer et al. (2021).

Most PET variables (i.e., radiation, wind speeds, surface pressure and actual vapour pressure) are only available on a monthly frequency in the CESM simulations, the near-surface air temperature is available on a daily frequency. Hence, we first need to verify whether such monthly-averaged data can be used to approximate daily PET rates, where we consider PET values derived from hourly-averaged data as the 'truth' (Singer et al., 2021). For this comparison we use ERA5 reanalysis data (Hersbach et al., 2020), from which we retained the same monthly-averaged and daily-averaged (from hourly averages) variables as available in CESM. The procedure of reconstructing daily-varying PET values (indicated by PET<sup>day</sup>) is presented in Appendix A, where we demonstrate that using daily-averaged data or monthly-averaged data gives reasonable PET rates (Figure A1).

The CESM provides monthly-averaged evaporation rates from the Community Land Model (Lawrence et al., 2011), but the land component exhibits various model biases when simulating these evaporation rates (Cheng et al., 2021). Instead, we calculate PET<sup>day</sup> for the CESM simulations, which has three advantages. First, the individual PET components in Equation (2) can be directly compared against ERA5 to identify model biases. For example, there is approximately 20% more net surface shortwave radiation over South and Central Europe in the PI<sub>18</sub> and PI<sub>18</sub> scenarios compared to ERA5 (Figure A2), leading to higher PET<sup>day</sup> rates (as shown in the Results). Second, the dominant drivers in PET<sup>day</sup> changes can be identified when comparing the different AMOC scenarios. Third, daily-averaged data can be used to reconstruct the water balance at a higher temporal resolution than the standard monthly frequency, which is useful for analysing the dry season length and intensity (see Section 2.3 below). A drawback of using PET<sup>day</sup> instead of the simulated evaporation rates is that the water balance is not closed. We acknowledge that CESM and different methodological choices introduce biases compared to ERA5; however, the main goal of this study is to analyse European hydroclimate responses under different AMOC regimes. We assess the impact of

AMOC collapse on hydroclimate by evaluating changes relative to the  $PI_{18}^{on}$  and  $PI_{45}^{on}$  scenarios, assuming that the biases remain constant between scenarios. This change signal is of primary interest. Therefore, from here onward, we will use  $PET^{day}$  for the analysis of the CESM simulations, resulting a surface water balance similar to before:  $W^{day} = P - PET^{day}$ . The evaporation responses in the CESM will be briefly discussed in the last Section 4.

We determine the Standardised Precipitation-Evapotranspiration Index (SPEI, Vicente-Serrano et al. (2010)), using the monthly-averaged water balance. We consider hydrological timescales by analysing SPEI-6, drought (wet) conditions are indicated by SPEI-6  $\leq -1$  (SPEI-6  $\geq 1$ ). We will use two variants of SPEI-6 calculations: first for each AMOC scenario using the PI<sub>18</sub> and PI<sub>45</sub> scenarios (referred to as SPEI-6<sup>ref</sup>), second for each AMOC scenario using its own climatology (referred to as SPEI-6). This second variant takes into account forced climatological mean changes but can be used to analyse forced changes in climate variability (van der Wiel and Bintanja, 2021). There are no notable variations in the second variations for PI<sup>off</sup>, RCP4.5 and RCP8.5, hence for these scenarios we only consider the SPEI-6<sup>ref</sup>.

#### 2.3 Dry Season





In the mid-latitudes, societal impacts of hydrological drought are most prominent during the growing season, as then local climatological PET rates exceed local climatological precipitation rates (Dullaart and van der Wiel, 2024). We define a 'dry season' by considering the Potential Precipitation Deficit (PPD). Note that the dry season may differ per region, hence we consider the local PPD. The PPD is set zero at the start of each calendar year (Dullaart and van der Wiel, 2024) and is obtained from the daily-averaged water balance:

$$PPD(t) = -\int_{1}^{t} W(t')dt'.$$
(3)

The negative PPD at the end of the calendar year indicates that, climatologically, precipitation exceeds potential evapotranspiration for most European regions in ERA5 (Figure 2a). Regions around the Mediterranean Sea and Black Sea have a positive PPD, indicating that in those regions there is a climatological precipitation deficit. Note that the actual evaporation rates are lower in these regions due to relatively low soil moisture content.

The local PPD at  $52^{\circ}$ N and  $5^{\circ}$ E (the Netherlands) is displayed in Figure 2b for ERA5. We consider this location as it can be compared with the measurement station 'De Bilt' and the local PPD in ERA5 agrees very well with observations (Dullaart and van der Wiel, 2024). This location is also of interest as it is situated in Northwestern Europe, a region that shows relatively large temperature responses under a collapsed AMOC (van Westen and Baatsen, 2025). We also display the median PPD<sup>day</sup> (derived from  $W^{day}$ ) for ERA5 in Figure 2b, which demonstrates that it is close to the median PPD with a difference of -16.5 mm by the end of the year. Note that this difference arises solely from PET<sup>day</sup>, and given that the yearly-integrated and local PET is 607 mm, the resulting error is only a few percent. In the main text we present the local PPD for the Netherlands and two other locations, situated in Sweden and Spain, are presented in Figure A3. The median PPD<sup>day</sup> is also close to the median PPD for these two locations in ERA5 (not shown) with a difference of +17.1 mm (Sweden) and +27.4 mm (Spain) by the end of the year.

To determine the local dry season we first retain the climatological median PPD and smooth it by a 15-day moving average (see inset Figure 2b). The dry season length is marked by the local minimum (here on 31 March) and local maximum (here on 4 August, length 127 days), the difference in PPD between these dates then quantifies the dry season intensity (here 85 mm). The dry season length needs to be at least 15 days long (i.e., moving average window length), otherwise the dry season length and its intensity are set to zero. Spatial patterns of the dry season length and dry season intensity are displayed in Figure 2c and Figure 2d, respectively. Some regions around the Mediterranean Sea have a dry season that spans almost the entire year, whereas regions in Scandinavia and the Alps have no notable dry season. The dry season length and intensity are not sensitive to slight variations in the moving average window length.

## 2.4 Atmospheric Circulation Regimes

A substantially weakened AMOC induces an anomalous anticyclonic atmospheric circulation over Europe (Orihuela-Pinto et al., 2022). Such an anomalous pattern could favour certain circulation regimes such as atmospheric blocking regimes. These blocking regimes are of particular interest as they induce persistent (i.e., few days) drier meteorological conditions over Europa (Michel et al., 2023) on top of the AMOC-induced changes.

To quantify the atmospheric circulation regimes, we followed the procedure outlined in Falkena et al. (2020) to detect different atmospheric circulation regimes using a k-means clustering algorithm. In addition to the standard k-means method, this approach includes a time-regularisation to identify the persistent regime signal without the need for low-pass filtering. First, we retained the daily-averaged mean sea-level pressure (MSLP) over the region between  $90^{\circ}\text{W} - 30^{\circ}\text{E}$  and  $20^{\circ}\text{N} - 80^{\circ}\text{N}$  and growing season (April – September, 183 days). Next, we subtracted the daily climatology to remove the annual cycle and the anomalies are then normalised to their surface area. Finally, we used k = 6 different clusters and assumed an average regime duration of 6.5 days (resulting in C = 5800). This 6.5 days was the typical (winter) regime lifetime found in observations (Falkena et al., 2020). The k-means clustering was repeated 100 times (with different initialisation conditions) and we selected the best averaged clustering functional (i.e., lowest L) (Franzke et al., 2009). Note that in most studies the daily-averaged 500 hPa geopotential height fields are used for identifying atmospheric circulation regimes as they are less impacted by surface variability, but these fields were not stored for the CESM simulations. The 500 hPa geopotential height anomalies induce comparable patterns for MSLP anomalies (Michel et al., 2023), meaning that the k-means clustering can still identify different regimes using daily-averaged MSLP fields. For more details on the k-means clustering algorithm and sensitivity experiments, we refer to Falkena et al. (2020).

#### 200 3 Results



This results section starts with the climatological PPD throughout the year in the eight AMOC scenarios, which are presented in section 3.1. Next in section 3.2, we analyse seasonal PPD changes by analysing the dry season length and intensity, together with a physical explanation of the drivers of PPD changes. Section 3.3 presents the drought extremes using SPEI-6. The final section 3.4 discusses the responses in the atmospheric circulation regimes and their associated precipitation patterns.

Figure 2. (a): The climatological potential precipitation deficit (PPD) at the end of the year for ERA5 (1981 – 2023). The PPD was determined using hourly-averaged PET and precipitation rates. Here we show the median PPD over the available 43-year period. (b): The local PPD at  $52^{\circ}$ N and  $5^{\circ}$ E (the Netherlands, diamond marker in panel a). The inset shows the dry season, which is derived from the climatological median PPD smoothed with a 15-day moving average. The climatological dry season at  $52^{\circ}$ N and  $5^{\circ}$ E starts on 31 March (PPD = -127 mm) and ends on 4 August (PPD = -42 mm), with a dry season intensity of 85 mm. The median PPD<sup>day</sup> is also shown (blue dashed curve). Spatial patterns of (c): the dry season length and (d): the dry season intensity.

# 3.1 The Climatological Potential Precipitation Deficit (PPD)





We start by analysing the PPD in the eight different CESM simulations (cf. Table 1) and obtained PPD<sup>day</sup> by reconstructing  $W^{\text{day}}$  using the daily-averaged precipitation rates and PET<sup>day</sup> over the 100-year periods. We determined the PET<sup>day</sup> also over water surfaces, as it convenient for the interpretation for the regional responses and the horizontal atmospheric resolution of the CESM is coarser (2°) compared to that of ERA5 (0.25°).

The climatological PPD<sup>day</sup> at the end of the year, together with the local PPD<sup>day</sup> in the Netherlands, are presented in Figure 3. The spatial patterns in the PPD<sup>day</sup> at the end of the year for the two PI<sup>on</sup> scenarios reasonably agree with the ERA5 (Figures 2a,c), for example Northern Europe has negative PPD<sup>day</sup> values and the opposite is true for Southern Europe. There are, however, regions in the PI<sup>on</sup> scenarios that are positively biased compared to ERA5. These biases mainly develop during the dry season (e.g., see local PPD<sup>day</sup> in Figures 3b,d) and are attributed to two factors. The first factor is the lower precipitation rates in the PI<sup>on</sup> scenarios compared to ERA5 during the growing season (Figure A2). The second factor is the higher PET<sup>day</sup> rates in CESM compared to ERA5, which is mainly related to more (about 20%) net surface shortwave radiation.

There is a persistent PPD<sup>day</sup> end of the year increase over most land surfaces for the PI<sup>off</sup>, RCP4.5 and RCP8.5 compared to their PI<sup>on</sup> scenario. Such PPD<sup>day</sup> responses were expected given that the yearly-averaged precipitation rates mainly reduce under the different scenarios (Figures 1c-j). Precipitation alone is not able to explain all the spatial (e.g., south – north) PPD<sup>day</sup> variations and larger PPD<sup>day</sup> responses are found under RCP8.5 than in RCP4.5. The latter suggests a prominent temperature contribution in the PET<sup>day</sup> responses, as PET is strongly dependent on the near-surface temperature. A part of this response is already shown for the PPD<sup>day</sup> in the Netherlands (Figure 3), where the dry season intensity increases under the climate change scenarios compared to their PI<sup>on</sup>. For the PI<sup>off</sup> scenarios, the local dry season intensity slightly decreases, which is likely related to land-ocean exchange, which is highly relevant for the Netherlands. For more continental locations, such as South Sweden (relatively short dry season) and North Spain (relatively long dry season), the dry season intensity increases for all scenarios compared to their PI<sup>on</sup> scenario (Figure A3).

The most interesting comparison is between RCP4. $5_{18}^{\rm on}$  and RCP4. $5_{45}^{\rm off}$ , where the scenarios differ in their AMOC regime (Figures 3i-l). The PPD<sup>day</sup> responses in RCP4. $5_{18}^{\rm on}$  are exacerbated under the collapsed AMOC in RCP4. $5_{45}^{\rm off}$ . For example for the Netherlands, the PPD<sup>day</sup> at the end of the year increases from -133 mm (RCP4. $5_{18}^{\rm on}$ ) to -62 mm (RCP4. $5_{45}^{\rm off}$ ), a difference of 71 mm. The RCP4. $5_{18}^{\rm on}$  also shows lower PPD<sup>day</sup> end of the year differences over Northwestern Europe compared to PI<sup>on</sup> (Figure 3i), whereas RCP4. $5_{45}^{\rm off}$  shows larger PPD<sup>day</sup> end of the year differences over almost all land surfaces (Figure 3k). This wetting response in RCP4. $5_{18}^{\rm on}$  is attributed to enhanced precipitation rates over Northwestern Europe during October to December (not shown). For example for the PPD<sup>day</sup> in the Netherlands, the amplitude of the maximum PPD<sup>day</sup> in September is very similar between the PI<sup>on</sup><sub>18</sub> and RCP4. $5_{18}^{\rm on}$  scenarios (black and blue curves in Figure 3j). Thereafter, the PPD<sup>day</sup> declines faster in RCP4. $5_{18}^{\rm on}$  than in PI<sup>on</sup><sub>18</sub>, meaning relatively wetter conditions for RCP4. $5_{18}^{\rm on}$ . The enhanced precipitation responses are likely linked to higher SSTs under climate change.

**Figure 3.** (First and third column): The climatological potential precipitation deficit (PPD<sup>day</sup>, median) at the end of the year. For the PI<sup>off</sup>, RCP4.5 and RCP8.5 scenarios, the PPD<sup>day</sup> are displayed as the difference compared to their PI<sup>on</sup> scenario. (Second and fourth column): The PPD<sup>day</sup> at 52°N and 5°E (the Netherlands), where the horizontal bar indicates the climatological dry season length and intensity. For the PI<sup>on</sup> scenarios, the median PPD for ERA5 is also displayed. For PI<sup>off</sup>, RCP4.5 and RCP8.5, the median PPD<sup>day</sup> for the PI<sup>on</sup> scenario is displayed.

The local PPD<sup>day</sup> end of the year differs by 71 mm when comparing the RCP4. $5_{18}^{\rm on}$  and RCP4. $5_{45}^{\rm off}$  scenarios. Their dry season intensity differs by 66 mm, which explain most of this PPD<sup>day</sup> end of the year response, suggesting an important contribution of dry season changes. These dry season responses will be explored in the following section.

# 240 3.2 Dry Season Precipitation and PET Responses







The results from the previous section show profound changes in the European hydroclimate under the different AMOC scenarios. In this section we analyse the dry season responses and its drivers. The dry season length and intensity are shown in Figure 4. Similar to the PPD<sup>day</sup> at the end of the year, most European land surfaces display an increase in the dry season intensity under all scenarios (compared to PI<sup>on</sup>) with the largest increase under the climate change simulations. The responses in the dry season intensity are larger for RCP4.5<sup>on</sup><sub>18</sub> than for RCP4.5<sup>off</sup><sub>45</sub>, and larger for RCP8.5 than for RCP4.5. Southern Europe shows the largest changes in dry season intensity.

The dry season over the Netherlands (Figure 3) starts 2-3 weeks later and its intensity slightly drops by 5 to 20 mm (-2 to -7%) in PI<sup>off</sup>. As was argued before, this reduced dry season intensity is likely related to land-ocean exchanges and for continental locations the dry season intensities increase under all scenarios (Figure A3). Nevertheless, for the two RCP4.5 scenarios we find an increase in the dry season intensity compared to their PI<sup>on</sup>, they increase by 8 % (RCP4.5<sup>on</sup><sub>18</sub>) and 28 % (RCP4.5<sup>off</sup><sub>45</sub>), a factor of 3.5 difference in their relative increase. This demonstrates again the exacerbating effects of an AMOC collapse. The largest increase for dry season length and intensity are found for the two RCP8.5 scenarios, the latter increases by about 60 %. Even more striking differences are found for the local dry season in Sweden and Spain (Figure A3). The dry season increases by 54% (40%) in RCP4.5<sup>on</sup><sub>18</sub> and by 72% (60%) in RCP4.5<sup>off</sup><sub>45</sub> for Sweden (Spain), and for the two RCP8.5 scenarios this is at least a factor of 2 for both locations.

The drivers of dry season changes can be understood by decomposing the PPD<sup>day</sup> into its precipitation and PET<sup>day</sup> contributions. Between regions the dry season length and period vary (Figures 4a,c) and hence we here consider a 'fixed' dry season between April and September, often referred to as the growing season. The growing season is characterised by relatively large PET rates because of higher temperatures and greater solar irradiance compared to the winter period (Dullaart and van der Wiel, 2024). The responses over the growing season, as presented in Figure 5, show that the precipitation responses (first and third column) are similar in all the collapsed AMOC scenarios. The RCP4.5<sup>on</sup><sub>18</sub> is, again, the exception here and shows a relatively small precipitation increase over Central and Northern Europe. These results indicate that an AMOC collapse contributes to a greater dry season intensity through reduced precipitation rates, given that the PET<sup>day</sup> rates are somewhat similar between RCP4.5<sup>on</sup><sub>18</sub> and RCP4.5<sup>off</sup><sub>45</sub>.

The precipitation responses over the growing season are not able to explain meridional dry season differences between Southern and Northern Europe, which suggests a prominent role for PET<sup>day</sup>. There are indeed meridional differences in the PET<sup>day</sup> responses (Figure 5, second and fourth column). For both PI<sup>off</sup> scenarios, the PET<sup>day</sup> responses are the opposite compared to the precipitation responses. The PI<sup>off</sup> scenarios have lower temperatures compared to PI<sup>on</sup> (van Westen and Baatsen, 2025) and hence reduce the PET<sup>day</sup> rates. These opposing precipitation and PET<sup>day</sup> responses explain the relatively limited responses in dry season length and intensity in PI<sup>off</sup> (Figure 4e-h). For all climate change scenarios the PET<sup>day</sup> increases

**Figure 4.** (First and third column): The dry season length and (second and fourth column): the dry season intensity. For the PI<sup>off</sup>, RCP4.5 and RCP8.5 scenarios, the dry season length and intensity are displayed as the difference compared to their PI<sup>on</sup> scenario.

Figure 5. (First and third column): The precipitation rates (colours) and mean sea-level pressures (contours) during the growing season (April – September). (Second and fourth column): the PET<sup>day</sup> rates during the growing season (April – September). For the PI<sup>off</sup>, RCP4.5 and RCP8.5 scenarios, the precipitation rates, mean sea-level pressures and PET<sup>day</sup> rates are displayed as the difference compared to their PI<sup>on</sup> scenario. The markers indicate non-significant ( $p \ge 0.05$ , two-sided Welch's t-test) precipitation and PET<sup>day</sup> differences.

under the higher atmospheric temperatures during the growing season, which is consistent with the intensification of the dry season. Southern Europe warms relatively stronger under climate change and AMOC collapse, correspondingly the largest PET<sup>day</sup> responses are found there. This relatively strong warming can be attributed to soil moisture depletion, which enhances the sensible heat fluxes while reducing the latent heat fluxes. Note that some parts of Northwestern Europe also cool under RCP4.5 $_{45}^{\text{off}}$  (van Westen and Baatsen, 2025).

The PET<sup>day</sup> responses are largely driven by temperature changes under the different AMOC scenarios. However, changes in wind speed and surface radiation may also contribute to PET<sup>day</sup> responses. These contributions can be isolated by determining PET<sup>day</sup> for the PI<sup>on</sup> and only modifying a single PET<sup>day</sup> variable. This variable is then obtained from the PI<sup>off</sup>, RCP4.5 or RCP8.5 scenario. For each year in PI<sup>on</sup>, we combined all the 100 years from one of the other scenarios, effectively determining 10,000 different PET<sup>day</sup> rates for the growing season. This procedure was done for each variable contributing to PET<sup>day</sup> (Equation (2)) to obtain their isolated response. Note that the near-surface temperatures ( $T_a$ ) and dew-point temperature ( $T_{dew}$ ) are strongly related and induce the opposite response on PET<sup>day</sup> changes. Hence for the temperature responses we consider the combined  $T_a$  and  $T_{dew}$  changes, where  $T_a$  is mostly explaining the sign of the PET<sup>day</sup> changes.

The two most dominant contributions in the PET<sup>day</sup> changes are temperature ( $\Delta T_a$  &  $\Delta T_{\rm dew}$ ) and net surface radiation ( $\Delta R_n$ ), which are displayed in Figure 6. It is clear that the temperature responses explain most of the PET<sup>day</sup> changes (compare to Figure 5) and the other contributions (e.g.,  $\Delta R_n$ ) are much smaller. There are regions that show significant PET<sup>day</sup> responses under  $\Delta R_n$  and these regions appear to overlap with the mean sea-level pressure anomaly patterns (see contours in Figure 5). For the scenarios in which the AMOC collapses, anomalous high pressure regions are found near Northwestern Europe. This anomalous patterns reduces cloud cover and enhances the net surface radiation through a larger (incoming) shortwave contribution. The mean sea-level pressures decrease under RCP4.5<sup>on</sup><sub>18</sub>, showing again an opposite response compared to RCP4.5<sup>off</sup><sub>45</sub>.

## 3.3 Drought Extremes




The previous sections showed climatologically drier conditions over Europe, based on these responses we expect more drought extremes. These extremes are quantified by SPEI-6  $\leq -1$  (drought) and SPEI-6  $\geq 1$  (wet conditions), which have a probability of about 17 % by definition and are shown in Figures 7a-d. All scenarios of PI<sup>off</sup>, RCP4.5 and RCP8.5 result in higher probabilities of drought and lower probabilities of wet conditions compared to their PI<sup>on</sup> (Figures 7e-p). For the RCP4.5<sup>on</sup><sub>18</sub>, the SPEI-6<sup>ref</sup> changes are smaller than in the RCP4.5<sup>off</sup><sub>45</sub>, indicating that an AMOC collapse further exacerbates the projected shifts to increased drought over Europe (Cook et al., 2020). Regions around the Mediterranean show again the largest changes, which are most pronounced for the two RCP8.5 scenarios. These SPEI-6<sup>ref</sup> responses align well with the results from the previous sections.

There is a persistent response over all calendar months (indicated by the markers in Figure 7) in the wet conditions over most European land surfaces in the two PI<sup>off</sup> scenarios, where extreme wet conditions become less likely. This can be attributed to reduced precipitation over Europe under a collapsed AMOC and explains the homogeneous decline in (extreme) wet conditions in PI<sup>off</sup>. Such a response is also found when comparing RCP4.5<sup>on</sup><sub>18</sub> (Figure 7j) and RCP4.5<sup>off</sup><sub>45</sub> (Figure 7l), where the

Figure 6. (a & b): The PET<sup>day</sup> rates during the growing season (April – September) for PI<sup>on</sup> . (c – n): The PET<sup>day</sup> differences during the growing season (April – September) for PI<sup>off</sup>, RCP4.5 and RCP8.5. The PET<sup>day</sup> responses are decomposed into a temperature contribution (i.e.,  $\Delta T_a$  &  $\Delta T_{\rm dew}$ , first and third column) and net surface radiation contribution (i.e.,  $\Delta R_n$ , second and fourth column). The markers indicate non-significant ( $p \ge 0.05$ , two-sided Welch's t-test) differences.

**Figure 7.** (First and third column): The probability of drought for SPEI-6 and SPEI-6<sup>ref</sup> over the 100-year periods. (Second and fourth column): The probability of wet conditions for SPEI and SPEI-6<sup>ref</sup> over the 100-year periods. For the PI<sup>off</sup>, RCP4.5 and RCP8.5 scenarios, the change in probabilities are displayed as the ratio (e.g.,  $R = \frac{\text{Scenario}}{\text{PI}^{\text{on}}}$ ) compared to their PI<sup>off</sup>. The markers in panels e – p indicate that all 12 calendar months have the same sign of their response (either R > 1 or R < 1).

latter scenario mainly shows less (extreme) wet conditions under a collapsed AMOC. Some regions show seasonally opposing SPEI-6<sup>ref</sup> responses (indicated by the absence of markers in Figure 7), which are mainly found over North(west)ern Europe. For example, for Northwestern Europe and under the RCP8.5 scenarios, the winter and early spring have wetter SPEI-6<sup>ref</sup> conditions, while the dry season has drier SPEI-6<sup>ref</sup> conditions (cf. Figure 3). In summary, the isolated AMOC-induced responses substantially reduce the wet conditions and increase drought occurrence. An AMOC collapse in combination with higher temperatures under climate change then mainly influences the drought extremes.

## 3.4 Dry Season Atmospheric Circulation Regimes

335

So far we have analysed the hydroclimate responses on yearly and seasonal timescales. In this section we present results on European atmospheric circulation regimes that usually last a few days to weeks (i.e., sub-monthly timescale). Two MSLP anomaly patterns from the k-means clustering are shown in Figure 8, together with their associated precipitation anomalies.

These specific two clusters are shown because they have maximum MSLPs (i.e., blockings) over Northwestern Europe and induce below-average precipitation rates over Europe, they are also the clusters with the highest frequency (~ 20%). The remaining clusters are shown in Figure A4 (clusters 3 & 4) and Figure A5 (clusters 5 & 6). Keep in mind that the MSLP and precipitation anomalies are with respect to the scenario background state. The spatial patterns of the atmospheric circulation regimes remain robust when comparing the different AMOC scenarios, though there are small displacements in the maximum MSLP location and variations in the frequency of each regime. The different atmospheric circulation regimes during the growing season appear to be resilient under different AMOC scenarios. The only exception is cluster 6, which shows more variety when comparing the different AMOC scenarios, which is left for future analysis.

The induced precipitation anomaly from cluster 1 and 2 can be determined as their weighted sum, and specifically for the PI<sup>on</sup> scenario:

325 
$$P_{1,2}^{\text{ref}} = \frac{f_1^{\text{ref}} P_1^{\text{ref}} + f_2^{\text{ref}} P_2^{\text{ref}}}{f_1^{\text{ref}} + f_2^{\text{ref}}},$$
 (4)

with  $f_i^{\text{ref}}$  and  $P_i^{\text{ref}}$  the cluster occurrence frequency and precipitation anomaly, respectively, for cluster i (= 1,2) and PI $^{\text{on}}$ . For the PI $^{\text{off}}$ , RCP4.5 and RCP8.5, a similar expression as in (4) can be used, however it is more relevant to analyse the precipitation anomalies as:

$$P_{1,2} = \frac{f_1 P_1 + f_2 P_2}{f_1^{\text{ref}} + f_2^{\text{ref}}}.$$
 (5)

The precipitation anomalies for the PI<sup>off</sup>, RCP4.5 and RCP8.5 are now weighted by the PI<sup>on</sup> to take any frequency variations into consideration. As the patterns in  $P_{1,2}$  are somewhat similar to  $P_{1,2}^{ref}$ , we display the differences compared to their PI<sup>on</sup> scenario (i.e.  $\Delta P_{1,2} = P_{1,2} - P_{1,2}^{ref}$ ), which are shown in Figure 9.

The two clusters induce negative precipitation anomalies over Northwestern Europe for all AMOC scenarios. The differences in the precipitation anomalies,  $\Delta P_{1,2}$ , are relatively small over the European continent. Although it appears that these two atmospheric regimes become effectively wetter over the European continent compared to PI<sup>on</sup>. A westward displacement of the MSLP maximum and more frequent cluster 1 for RCP4.5 $_{45}^{\rm off}$  (Figure 8i) induce drier conditions over the North Atlantic

**Figure 8.** The area-normalised MSLP anomaly patterns in the growing season (colours) from the *k*-means clustering algorithm and their frequency for the different AMOC scenarios. The circled markers indicate the maximum (red) and minimum (blue) in the MSLP anomaly patterns. The contours show the associated precipitation anomalies (not normalised with area) for the given cluster.

Ocean. In summary, the typical weather regimes do not change much in their overall pattern nor frequency, but small spatial variations may results in a slightly different precipitation anomaly patterns compared to the PI<sup>on</sup> scenario.

## 4 Discussion

By analysing daily-averaged precipitation rates and reconstructed daily potential evapotranspiration rates in the Community Earth System Model (CESM), we obtained daily water balances which were used to analyse the climatological potential precipitation deficit, mean dry season responses, and changes in the frequency of drought extremes. The PI<sub>18</sub> and PI<sub>45</sub> were used as reference (i.e., the AMOC on state) and for comparison with the collapsed AMOC states (PI<sub>18</sub>, PI<sub>45</sub>) and climate change scenarios (RCP4.5<sub>18</sub>, RCP8.5<sub>18</sub>, RCP4.5<sub>45</sub>, RCP8.5<sub>45</sub>). In the PI<sub>18</sub> and PI<sub>45</sub>, both precipitation rates and PET<sup>day</sup> rates decrease, where the former is the most dominant response resulting in drier conditions. The PET<sup>day</sup> rates decline as the European climate cools under these scenarios (van Westen and Baatsen, 2025), which partly offsets the reduced precipitation rates. The RCP8.5<sub>18</sub> and RCP8.5<sub>45</sub> both showed an AMOC collapse under the high emission scenario and have a similar

**Figure 9.** The precipitation anomaly patterns from the k-means clustering algorithm for  $P_{1,2}^{ref}$  (PI<sup>on</sup>) and  $\Delta P_{1,2}$  (PI<sup>off</sup>, RCP4.5 and RCP8.5).

precipitation response as  $PI_{18}^{off}$  and  $PI_{45}^{off}$ . The PET<sup>day</sup> rates in RCP8.5<sup>off</sup><sub>18</sub> and RCP8.5<sup>off</sup><sub>45</sub>, however, are strongly increasing driven by higher atmospheric temperatures due to climate change, resulting in a more intense dry season and drought extremes.




The most interesting comparison is made between the two RCP4.5 scenarios as they differ in their AMOC regime. For the RCP4.5 scenarios, the global climate and European climate warm and this scenario represents the isolated hydroclimate responses under anthropogenic climate change with AMOC strengths close to present-day values (Srokosz and Bryden, 2015). The RCP4.5 scenarios are combination of both anthropogenic climate change and a collapsed AMOC. The 'standard' projected increases in dry season intensity and drought extremes under climate change (e.g., Cook et al. (2020); van der Wiel et al. (2023)) are exacerbated under an AMOC collapse, consistent with previous regional analyses on AMOC tipping behaviour (Ritchie et al., 2020; Laybourn et al., 2024). This highlights the importance of considering the potential of AMOC tipping behaviour in studies and decision making on hydroclimatic topics.

The systematic analysis and decomposition of precipitation and PET<sup>day</sup> responses reveal a drying response over Europe. However, the 'actual' evaporation rates are constrained by the available soil moisture content. A reduction in precipitation (P) due to an AMOC collapse will, in turn, lead to a reduction in evaporation (E). These opposing responses between precipitation and evaporation result in a limited change in E minus P (Figure 10). The overall drying response under an AMOC collapse

remains robust over Central Europe when analysing E-P. However, in Southern Europe, where soil moisture content is already low, E-P decreases (i.e., wetter) under an AMOC collapse. The AMOC-induced drier conditions are less pronounced in E-P and hence it is more useful to analyse the water balances (with P-PET $^{day}$ ).

Figure 10. (First and third column): The evaporation rates during the growing season (April – September). (Second and fourth column): the evaporation minus precipitation (E-P) rates during the growing season (April – September). For the PI<sup>off</sup>, RCP4.5 and RCP8.5 scenarios, the precipitation rates and E-P rates are displayed as the difference compared to their PI<sup>off</sup> scenario. The markers indicate non-significant  $(p \ge 0.05$ , two-sided Welch's t-test) differences.

As was argued in van Westen and Baatsen (2025), the CESM version used here has different biases compared to reanalysis (ERA5) data. There is for example less precipitation during the growing season (Figure A2), while having more precipita-


tion during the winter season (not shown). This is a typical bias found in the models participating in the Coupled Model Intercomparison Project phase 6 (CMIP6) (Osso et al., 2023). The precipitation bias during the growing season could induce higher near-surface temperatures, as sensible heat fluxes are favoured over latent heat fluxes under relatively dry conditions. These higher near-surface temperatures then enhance PET<sup>day</sup> rates. However, the PET<sup>day</sup> was mostly influenced by much more (+20%) solar radiation over Europe in PI<sup>on</sup><sub>18</sub> and PI<sup>on</sup><sub>45</sub> compared to reanalysis. This solar radiation bias suggests a poor cloud representation in CESM, which is a well-documented climate model bias (Wild et al., 1996; Soden and Held, 2006; Chen et al., 2022). Although the CESM version used here shows persistent hydroclimate biases, we assumed that those biases remain constant when comparing the different AMOC scenario. Note that climate models are tuned under a strong AMOC state and hence this assumption is likely not valid for the collapsed AMOC state. We do expect that the AMOC-induced changes are (much) larger than variations in climate model biases, but this cannot be tested.

Part of these biases can be attributed to the 2° atmospheric horizontal resolution in our CESM simulation. This resolution allows to resolve the synoptic scale and mesoscale features are parameterised. Enhancing the atmospheric horizontal resolution to 0.25° does not substantially improve European precipitation biases in the CESM (Chang et al., 2020), and possibly an even higher resolution is required to resolve all relevant (sub)mesoscale processes (Hentgen et al., 2019). A higher horizontal resolution, however, can improve the representation of atmospheric blocking regimes (Michel et al., 2023). For the latter, we found no substantial responses in the atmospheric circulation regimes under the different AMOC regimes.

Another point to consider is the imposed freshwater flux forcing  $F_H$ , to obtain a more sensitive AMOC under climate change, which essentially acts as an AMOC bias correction as well. The latest generation climate models have an overly stable AMOC and likely underestimate the risk of AMOC tipping under climate change (Van Westen and Dijkstra, 2024; Vanderborght et al., 2025). Although this bias correction is far from ideal, it allows us to analyse the two RCP4.5 scenarios where the AMOC-induced responses were most striking. It would be interesting to conduct a similar hydroclimate analysis using other climate models that have a substantially weaker AMOC strengths under hosing and/or climate change (Jackson et al., 2023; Romanou et al., 2023; Saini et al., 2025). At least for European precipitation, the CESM results are comparable with that of CLIMBER-2 (Rahmstorf and Ganopolski, 1999), HadCM3 (Vellinga and Wood, 2002), EC-Earth3 (Bellomo et al., 2023) and HadGEM3 (Ritchie et al., 2020). Since changes in PET<sup>day</sup> are primarily driven by near-surface temperatures, we expect a robust drying response during the growing season across climate models that simulate an AMOC collapse under climate change. Such a model intercomparison analysis would aid in improving drought projections under an AMOC collapse scenario, given that CESM exhibits substantial biases over several European regions.

## **5 Summary**

In this study, we presented results on the European hydroclimate responses for eight scenarios with different combinations of AMOC strength (with and without collapse) and anthropogenic climate change (pre-industrial, RCP4.5 and RCP8.5). The analysis focussed on the European continent, a region that shows relatively large responses in its climate mean state under a collapsing AMOC (van Westen et al., 2024b, 2025b). The aim of this study was to provide a quantitative assessment of the

balance between precipitation and potential evapotranspiration changes under different AMOC regimes in the CESM. The results indicate that the annual mean precipitation and the precipitation over the growing season (April – September) decline under a collapsed AMOC. The growing season is expected to have more droughts under climate change (Cook et al., 2020; van der Wiel et al., 2023) and an AMOC collapse exacerbates this drying response.

A more intense dry season and more droughts can have severe societal and ecological impacts (Ritchie et al., 2020; van der Wiel et al., 2023; Laybourn et al., 2024; Lee et al., 2025; van Thienen et al., 2025). Given the societal and ecological relevance of the here noted impacts, hydroclimate projections for the (far) future need to consider the exacerbated effects of a potential weaker or fully-collapsed AMOC state. Note that we do not expect that the AMOC reaches a fully-collapsed state before 2100, given that it takes more than 100 years to reach a substantially weaker AMOC state (van Westen et al., 2024b). If the AMOC begins to collapse, transient responses are expected to dominate first and the presented drier hydroclimate conditions are expected (far) beyond 2100.

Code and data availability. All model output and code to generate the results are available at: https://doi.org/10.5281/zenodo.16905376. The hourly-averaged PET in ERA5, which was converted to daily averages, are accessible at: https://doi.org/10.5523/bris.qb8ujazzda0s2aykkv0oq0ctp (Singer et al., 2021). The hourly-averaged ERA5 data (used for daily-averaged temperatures) can be accessed at: https://doi.org/10.24381/cds.adbb2d47, the monthly-averaged ERA5 data is found at: https://doi.org/10.24381/cds.f17050d7.

## 415 Appendix A: Derivation of monthly-varying and daily-varying PET rates

To obtain a monthly-varying PET, the first step is to split the PET into a daytime and nighttime contribution to account for the G dependency:

$$PET^{\text{daytime}} = \frac{0.408\Delta(0.9R_n^{\text{daytime}}) + \gamma\left(\frac{37}{T_a + 273.15}\right)u_2(e_s - e_a)}{\Delta + \gamma(1 + 0.34u_2)}$$
(A1)

and

PET<sup>nighttime</sup> = 
$$\frac{0.408\Delta(0.5R_n^{\text{nighttime}}) + \gamma\left(\frac{37}{T_a + 273.15}\right)u_2(e_s - e_a)}{\Delta + \gamma(1 + 0.34u_2)}.$$
 (A2)

The daytime and nightime net surface radiation are defined as:

$$R_n^{\text{daytime}} = R_s^{\text{daytime}} - R_l, \tag{A3}$$

$$R_n^{\text{nightime}} = 0 - R_l, \tag{A4}$$

with  $R_s^{\text{daytime}}$  the net shortwave radiation at the surface during daytime, and  $R_l$  the net longwave radiation at the surface.

All variables in relation (A1) through (A4) are determined using monthly-averaged data. The monthly-averaged net shortwave radiation at the surface (i.e.,  $R_s$ ) is biased to zero because of nighttime contributions and needs to be corrected using the day

length. For the day length calculation (e.g., see Sproul (2007)), we require the solar hour angle ( $\omega_0$ ), which is a function of the latitude ( $\phi$ ) and sun declination angle ( $\delta$ ):

$$\cos\omega_0 = -\tan\phi\tan\delta,\tag{A5}$$

$$\delta = 23.45^{\circ} \times \cos\left(\frac{d - 172}{365}360^{\circ}\right),$$
 (A6)

with d the day of the year ( $d=1 \rightarrow 1$  January, omitting leap years). The trigonometry functions and quantities are in degrees. The local sunrise (at z=0) is then at  $\tau_{\rm rise}=12-\frac{\omega_0}{15^\circ}$  hour and local sunset at  $\tau_{\rm set}=12+\frac{\omega_0}{15^\circ}$  hour. Note that we do not consider time corrections for the longitudinal coordinate and altitude variations, the latter hardly influences the results. Finally, the day length (in hours) is given by:

$$\tau = \tau_{\text{set}} - \tau_{\text{rise}} = 2\frac{\omega_0}{15^{\circ}},$$
 (A7)

We introduce the daytime scaling factor,  $f_{\tau}=\frac{24}{\tau}$ , to adjust the monthly-averaged net shortwave radiation  $(R_s)$ . For example, consider the local  $R_s=150~{\rm W~m^{-2}}$  at  $\phi=49.5^{\circ}{\rm N}$  for a random June, with the associated monthly-averaged  $\tau=16$  hours and  $f_{\tau}=1.5$ . The daytime net shortwave radiation is then  $R_s^{\rm daytime}=f_{\tau}R_s=225~{\rm W~m^{-2}}$ . Keep in mind that at the higher latitudes the day length can be zero (i.e., the polar nights), the net shortwave radiation is then by definition zero and we omit the daytime scaling factor in these cases.

The last step is to determine the local and monthly-averaged PET, indicated by PET<sup>month</sup>, and is calculated as:


$$PET^{month} = \frac{\tau}{24} PET^{daytime} + \frac{1-\tau}{24} PET^{nighttime}.$$
 (A8)

Instead of using the monthly-averaged temperatures, we can also use the daily-averaged temperatures in combination with the remaining monthly-averaged variables. We follow the same steps from (A1) through (A8), but then have a daily-varying PET<sup>daytime</sup> and PET<sup>nighttime</sup>, and we refer to this quantity as PET<sup>day</sup>. The advantage of PET<sup>day</sup> over PET<sup>month</sup> is that day-to-day fluctuations are partly represented, as PET is strongly dependent on temperature. More details on the calculations of PET<sup>day</sup> and PET<sup>month</sup> are provided in the openly-available Python codes.

Below in Figure A1, we present the PET comparison over the growing season (April – September), the annual PET comparison is available in the Zenodo repository. Differences from PET are relatively small in both PET<sup>day</sup> and PET<sup>month</sup> (Figures A1b,c). The area-weighted root-mean-square deviation over the shown land surfaces is 0.11 mm day<sup>-1</sup> for both PET<sup>day</sup> and PET<sup>month</sup>. There are relatively large PET<sup>day</sup> and PET<sup>month</sup> deviations over Scandinavia, which can partly attributed to the relatively low PET rates there. The hourly-averaged PET rates were converted to daily averages and monthly averages to determine the root-mean-square error (RMSE) for PET<sup>day</sup> and PET<sup>month</sup>, respectively (Figures A1d,e). For example, the local and daily RMSE was determined as:

RMSE = 
$$\sqrt{\frac{1}{T} \sum_{t=1}^{T} \left( \text{PET}^{\text{day}}(t) - \overline{\text{PET}(t)} \right)^2}$$
, (A9)

with T the number of days and  $\overline{PET}$  the daily-averaged PET (from hourly averages). The RMSE in PET<sup>month</sup> is substantially smaller than the RMSE in PET<sup>day</sup> and can be explained that the monthly-averaged PET is quite close to PET<sup>month</sup>. However,

comparing PET<sup>month</sup> to daily-varying PET (as is done for PET<sup>day</sup>) results in larger RMSE in PET<sup>month</sup> (not shown) than the RMSE for PET<sup>day</sup>. The daily temperature fluctuations are (partly) represented in PET<sup>day</sup> and hence closer to the daily-varying PET. We also determine the seasonally-integrated PET<sup>day</sup> and PET<sup>month</sup> at the end of the growing season:

$$PET^{\text{end}} = \int_{1 \text{ Apr}}^{30 \text{ Sep}} PET(t')dt', \tag{A10}$$

with the local RMSE at the end of the growing season given by:

$$RMSE^{end} = \sqrt{\frac{1}{Y} \sum_{t=1}^{Y} \left( PET^{day,end}(t) - PET^{end}(t) \right)^2},$$
(A11)

with Y the number of years (with a similar expression for PET<sup>month</sup>). The RMSE<sup>end</sup> are shown in Figures A1f,g and the differences are less than 30 mm over most land surfaces and end of growing season, boiling down to an error of a few percents as the seasonally-integrated PET is typically more than 550 mm.

In summary, the PET<sup>day</sup> rates may deviate from the daily-averaged PET and one must be careful with the interpretation of day-to-day PET<sup>day</sup>, but for longer time scales (weeks to months) the PET<sup>day</sup> is close to PET. We conclude that averaging hourly data gives reasonable PET rates in ERA5. This approach can then also be applied to global climate model output, where relevant climate variables are determined at a high frequency (typically 

**Figure A1.** (a): The hourly-averaged PET (i.e., truth) for the growing season (April – September) in ERA5 (1981 – 2023). (b & c): Similar to panel a, but now the PET<sup>day</sup> and PET<sup>month</sup> expressed as the relative difference from PET. (d & e): The root-mean-square error (RMSE) for PET<sup>day</sup> and PET<sup>month</sup> against the hourly-averaged PET. (f & g): The RMSE at the end of the growing season (RMSE<sup>end</sup>) by integrating hourly-averaged PET, PET<sup>day</sup> and PET<sup>month</sup> over the growing season.

Figure A2. The precipitation, near-surface (2-meter) temperatures, 10-meter wind speed, net surface shortwave radiation, and net surface radiation for ERA5 (1981 – 2023),  $PI_{18}^{on}$  and  $PI_{45}^{on}$ . The climate variables are determined over the growing season (April – September). For the  $PI_{45}^{on}$  scenarios, the climate variables are displayed as the difference compared to ERA5.

Figure A3. Similar to Figure 3, but now for  $60^{\circ}$ N and  $15^{\circ}$ E (Sweden) and  $42.5^{\circ}$ N and  $5^{\circ}$ W (Spain). Note the different vertical ranges between the two locations.

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

**Figure A4.** Similar to Figure 8, but now for clusters 3 and 4.

- Chang, P., Zhang, S., Danabasoglu, G., Yeager, S. G., Fu, H., Wang, H., Castruccio, F. S., Chen, Y., Edwards, J., Fu, D., et al.: An unprecedented set of high-resolution earth system simulations for understanding multiscale interactions in climate variability and change, Journal of Advances in Modeling Earth Systems, 12, e2020MS002 298, 2020.
  - Chen, G., Wang, W.-C., Bao, Q., and Li, J.: Evaluation of simulated cloud diurnal variation in CMIP6 climate models, Journal of Geophysical Research: Atmospheres, 127, e2021JD036 422, 2022.
- Cheng, Y., Huang, M., Zhu, B., Bisht, G., Zhou, T., Liu, Y., Song, F., and He, X.: Validation of the community land model version 5 over the contiguous United States (CONUS) using in situ and remote sensing data sets, Journal of Geophysical Research: Atmospheres, 126, e2020JD033 539, 2021.
  - Cook, B. I., Mankin, J. S., Marvel, K., Williams, A. P., Smerdon, J. E., and Anchukaitis, K. J.: Twenty-first century drought projections in the CMIP6 forcing scenarios, Earth's Future, 8, e2019EF001461, 2020.
- Dullaart, J. and van der Wiel, K.: Underestimation of meteorological drought intensity due to lengthening of the drought season with climate change, Environmental Research: Climate, 3, 041 004, 2024.
  - Falkena, S. K., de Wiljes, J., Weisheimer, A., and Shepherd, T. G.: Revisiting the identification of wintertime atmospheric circulation regimes in the Euro-Atlantic sector, Quarterly Journal of the Royal Meteorological Society, 146, 2801–2814, 2020.
  - Franzke, C., Horenko, I., Majda, A. J., and Klein, R.: Systematic metastable atmospheric regime identification in an AGCM, Journal of the Atmospheric Sciences, 66, 1997–2012, 2009.
- Hentgen, L., Ban, N., Kröner, N., Leutwyler, D., and Schär, C.: Clouds in convection-resolving climate simulations over Europe, Journal of Geophysical Research: Atmospheres, 124, 3849–3870, 2019.

**Figure A5.** Similar to Figure 8, but now for clusters 5 and 6.

510

- Hersbach, H., Bell, B., Berrisford, P., Hirahara, S., Horányi, A., Muñoz-Sabater, J., Nicolas, J., Peubey, C., Radu, R., Schepers, D., et al.: The ERA5 global reanalysis, Quarterly journal of the royal meteorological society, 146, 1999–2049, 2020.
- Ionita, M., Nagavciuc, V., Scholz, P., and Dima, M.: Long-term drought intensification over Europe driven by the weakening trend of the Atlantic Meridional Overturning Circulation, Journal of Hydrology: Regional Studies, 42, 101 176, 2022.
- Jackson, L. C., Kahana, R., Graham, T., Ringer, M. A., Woollings, T., Mecking, J. V., and Wood, R. A.: Global and European climate impacts of a slowdown of the AMOC in a high resolution GCM, Climate Dynamics, 45, 3299 3316, https://doi.org/10.1007/s00382-015-2540-2, 2015.
- Jackson, L. C., Alastrué de Asenjo, E., Bellomo, K., Danabasoglu, G., Haak, H., Hu, A., Jungclaus, J., Lee, W., Meccia, V. L., Saenko, O., et al.: Understanding AMOC stability: the North Atlantic hosing model intercomparison project, Geoscientific Model Development, 2022, 1–32, 2023.
  - Jacob, D., Goettel, H., Jungclaus, J., Muskulus, M., Podzun, R., and Marotzke, J.: Slowdown of the thermohaline circulation causes enhanced maritime climate influence and snow cover over Europe, Geophysical research letters, 32, 2005.
- Johns, W. E., Baringer, M. O., Beal, L., Cunningham, S., Kanzow, T., Bryden, H. L., Hirschi, J., Marotzke, J., Meinen, C., Shaw, B., et al.:

  Continuous, array-based estimates of Atlantic Ocean heat transport at 26.5 N, Journal of Climate, 24, 2429–2449, 2011.
  - Lawrence, D. M., Oleson, K. W., Flanner, M. G., Thornton, P. E., Swenson, S. C., Lawrence, P. J., Zeng, X., Yang, Z.-L., Levis, S., Sakaguchi, K., et al.: Parameterization improvements and functional and structural advances in version 4 of the Community Land Model, Journal of Advances in Modeling Earth Systems, 3, 2011.

- Laybourn, L., Abrams, J. F., Benton, D., Brown, K., Evans, J., Elliot, J., Swingedouw, D., Lenton, T. M., and Dyke, J. G.: The Blind Spot.

  Cascading Climate Impacts and Tipping Points Threaten National Security, The Institute for Public Policy Research, 2024.
  - Lee, J., Biemond, B., van Keulen, Daan, H. Y., van Westen, R. M., de Swart, H. E., Dijkstra, H. A., and Kranenburg, W. M.: Global increases in salt intrusion in estuaries under future environmental conditions, Accepted for Nature Communications, 2025.
  - Levermann, A., Griesel, A., Hofmann, M., Montoya, M., and Rahmstorf, S.: Dynamic sea level changes following changes in the thermohaline circulation, Climate Dynamics, 24, 347–354, 2005.
- 530 Meccia, V. L., Simolo, C., Bellomo, K., and Corti, S.: Extreme cold events in Europe under a reduced AMOC, Environmental Research Letters, 19, 014 054, 2024.
  - Michel, S. L., von der Heydt, A. S., van Westen, R. M., Baatsen, M. L., and Dijkstra, H. A.: Increased wintertime European atmospheric blocking frequencies in General Circulation Models with an eddy-permitting ocean, npj Climate and Atmospheric Science, 6, 50, 2023.
- Monteith, J. L.: Evaporation and environment, in: Symposia of the society for experimental biology, vol. 19, pp. 205–234, Cambridge University Press (CUP) Cambridge, 1965.
  - Orihuela-Pinto, B., England, M. H., and Taschetto, A. S.: Interbasin and interhemispheric impacts of a collapsed Atlantic Overturning Circulation, Nature Climate Change, 12, 558–565, 2022.
  - Osso, A., Craig, P., and Allan, R. P.: An assessment of CMIP6 climate signals and biases in temperature, precipitation and soil moisture over Europe, International Journal of Climatology, 43, 5698–5719, 2023.
- 540 Penman, H. L.: Natural evaporation from open water, bare soil and grass, Proceedings of the Royal Society of London. Series A. Mathematical and Physical Sciences, 193, 120–145, 1948.
  - Rahmstorf, S. and Ganopolski, A.: Long-term global warming scenarios computed with an efficient coupled climate model, Climatic change, 43, 353–367, 1999.
- Ritchie, P. D., Smith, G. S., Davis, K. J., Fezzi, C., Halleck-Vega, S., Harper, A. B., Boulton, C. A., Binner, A. R., Day, B. H., Gallego-Sala,
  A. V., et al.: Shifts in national land use and food production in Great Britain after a climate tipping point, Nature Food, 1, 76–83, 2020.
  - Romanou, A., Rind, D., Jonas, J., Miller, R., Kelley, M., Russell, G., Orbe, C., Nazarenko, L., Latto, R., and Schmidt, G. A.: Stochastic bifurcation of the North Atlantic circulation under a midrange future climate scenario with the NASA-GISS ModelE, Journal of Climate, 36, 6141–6161, 2023.
- Saini, H., Pontes, G., Brown, J. R., Drysdale, R. N., Du, Y., and Menviel, L.: Australasian hydroclimate response to the collapse of the atlantic meridional overturning circulation under pre-industrial and last interglacial climates, Paleoceanography and Paleoclimatology, 40, e2024PA004 967, 2025.
  - Singer, M. B., Asfaw, D. T., Rosolem, R., Cuthbert, M. O., Miralles, D. G., MacLeod, D., Quichimbo, E. A., and Michaelides, K.: Hourly potential evapotranspiration at 0.1 resolution for the global land surface from 1981-present, Scientific Data, 8, 224, 2021.
- Soden, B. J. and Held, I. M.: An assessment of climate feedbacks in coupled ocean–atmosphere models, Journal of climate, 19, 3354–3360, 2006.
  - Sproul, A. B.: Derivation of the solar geometric relationships using vector analysis, Renewable energy, 32, 1187–1205, 2007.
  - Srokosz, M. A. and Bryden, H. L.: Observing the Atlantic Meridional Overturning Circulation yields a decade of inevitable surprises., Science, 348, 1255 575 1255 575, https://doi.org/10.1126/science.1255575, 2015.
- van der Wiel, K. and Bintanja, R.: Contribution of climatic changes in mean and variability to monthly temperature and precipitation extremes, Communications Earth & Environment, 2, 1, 2021.

- van der Wiel, K., Batelaan, T. J., and Wanders, N.: Large increases of multi-year droughts in north-western Europe in a warmer climate, Climate Dynamics, 60, 1781–1800, 2023.
- van Thienen, P., ter Maat, H., and Stofberg, S.: Climate tipping points and their potential impact on drinking water supply planning and management in Europe, Cambridge Prisms: Water, 3, e3, 2025.
- van Westen, R. M. and Baatsen, M. L.: European temperature extremes under different AMOC scenarios in the community Earth system model, Geophysical Research Letters, 52, e2025GL114611, 2025.
  - van Westen, R. M. and Dijkstra, H. A.: Asymmetry of AMOC Hysteresis in a State-Of-The-Art Global Climate Model, Geophysical Research Letters, 50, e2023GL106 088, 2023.
- Van Westen, R. M. and Dijkstra, H. A.: Persistent climate model biases in the Atlantic Ocean's freshwater transport, Ocean Science, 20, 549–567, 2024.
  - van Westen, R. M., Jacques-Dumas, V., Boot, A. A., and Dijkstra, H. A.: The Role of Sea ice Insulation Effects on the Probability of AMOC Transitions, Journal of Climate, 2024a.
  - van Westen, R. M., Kliphuis, M., and Dijkstra, H. A.: Physics-based early warning signal shows that AMOC is on tipping course, Science advances, 10, eadk1189, 2024b.
- van Westen, R. M., Kliphuis, M., and Dijkstra, H. A.: Collapse of the Atlantic Meridional Overturning Circulation in a Strongly Eddying Ocean-Only Model, Geophysical Research Letters, 52, e2024GL114 532, 2025a.
  - van Westen, R. M., Vanderborght, E., Kliphuis, M., and Dijkstra, H. A.: Physics-based Indicators for the Onset of an AMOC Collapse under Climate Change, Journal of Geophysical Research: Oceans, 2025b.
- Vanderborght, E., van Westen, R. M., and Dijkstra, H. A.: Feedback processes causing an AMOC collapse in the community earth system model, Journal of Climate, 1, 2025.
  - Vellinga, M. and Wood, R. A.: Global climatic impacts of a collapse of the Atlantic thermohaline circulation, Climatic change, 54, 251–267, 2002.
  - Vicente-Serrano, S. M., Beguería, S., and López-Moreno, J. I.: A multiscalar drought index sensitive to global warming: the standardized precipitation evapotranspiration index, Journal of climate, 23, 1696–1718, 2010.
- Wild, M., Dümenil, L., and Schulz, J.-P.: Regional climate simulation with a high resolution GCM: surface hydrology, Climate Dynamics, 12, 755–774, 1996.