# Peer review of "Changing European Hydroclimate under a Collapsed AMOC in the Community Earth System Model"

_EGUsphere, 2025_

## Referee Comment (RC1)

**egusphere-2025-1440**

**1    Summary**

In the manuscript *Changing European Hydroclimate under a Collapsed AMOC in the Community Earth System Model*, eight model simulations are used to assess the impact of the AMOC strength and two different Representative Concentration Pathways (RCP4.5 and RCP8.5) on the European hydroclimate. The authors reconstruct daily potential evapotranspiration rates (PET) from CESM variables stored at monthly frequency. This allows them to diagnose the differences in dry season length and intensity under the different scenarios, as well as determining the dominant terms in the PET equation that are most sensitive to the changing climatic conditions in the eight simulations. The authors find that compared to the pre-industrial conditions (PI), nearly all scenarios show a general reduction in the potential precipitation deficit (PPD) over Europe, resulting in generally drier climate and an increase in drought extremes.

The eight scenarios allow for a comparison between the effects of the AMOC shutdown and the anthropogenic climate change on the European hydroclimate. Under PI conditions with reduced AMOC strength, the precipitation and PET rates decrease, and the latter dominates, resulting in dry conditions. In the RCP8.5 scenarios, the precipitation is reduced as well, but the PET rates increase, and the two effects combine, resulting in more intense and longer dry periods than in the AMOC off scenarios. Lastly, the RCP4.5 simulations provide the most important comparison between the European climate states under increased radiative fluxes with and without collapsed AMOC. These results show how AMOC collapse intensifies the dry conditions in Europe in the warming climate. The RCP4.5 scenario with collapsed AMOC shows significantly longer and more intense dry season in Southern and Central Europe compared to the AMOC on case, as well as significantly increased probability for drought.

The study presents a meaningful and important contribution to understanding the risks associated with the combined effect of climate change and AMOC tipping on the hydroclimate. As such, the article is a relevant addition to the Hydrology and Earth System Sciences (HESS) Journal, both in terms of scientific content and result significance.

**2    General comments**

The manuscript is well organized and the analysis is conducted to a high standard. Analyzing the PPD and its decomposition into precipitation and PET rates reveals the pathways driving changes in European hydroclimate across scenarios. The changes in precipitation are linked to mean sea-level pressures, and it is shown that the dominant factor affecting the PET is temperature. It is furthermore demonstrated that reconstructing $\mathrm{PET}^{\mathrm{day}}$ is important for understanding the scenario impact, as the $E - P$ signal is weaker than $P - PET$, since evaporation in the model is limited by the precipitation rate.

The reconstruction of $\mathrm{PET}^{\mathrm{day}}$ and $\mathrm{PET}^{\mathrm{month}}$ in Appendix A is well justified and described. However, comparing the RMSE of end-year PPD computed from true and reconstructed PET would be a more meaningful measure of the biases introduced with the upscaling method. Since the CESM PPD biases are significant compared to ERA5, it would be useful to know to which extent the PET reconstruction may be contributing to these biases, if at all.

The communication, especially in the results section, is lacking clarity and should be improved. First of all, the significance of the $k$-means analysis is unclear. The authors

should state clearly the significance of this analysis and how it fits with the previous results. More importantly, it takes a long time to realize that the focus for the reader should be the difference between the two RCP4.5 scenarios, where the role of the AMOC collapse in contributing to the hydroclimate changes under increased radiative forcing can be identified. This important result drowns in the overly detailed text of the section, and is also not clearly visually indicated on the figures. The manuscript clarity would improve if both the results and the figures were streamlined to better express the focus on the differences between the results from the two RCP4.5 scenarios.

The naming convention for the different scenarios greatly contributes to the lack of clarity and transparency in the text and on the figures. Instead of the cumbersome 'AMOC on ($F_H = 0.18$)', the authors could consider simply writing 'PI_LOW' or 'ON_LOW' to indicate the lower rate of freshwater hosing. It is not crucial for the reader to know the exact $F_H$ value throughout the text. It is also not obvious how the simulations in the manuscript are branched off from the simulations with $F_H = 0.66$ Sv (ll. 65-69). This can be clarified. Additionally, the figure titles and labels are very small, and increasing the font size would improve the ease of understanding.

Lastly, the discussion about model biases is lacking. The assumption that CESM biases with respect to ERA5 stay constant is highly unlikely to hold (ll. 111-113). It would be great if the authors could include more discussion about how the significant CESM biases may affect the results.

**3  Minor comments**

It is surprising that the Netherlands is chosen as the reference location to show the local changes in PPD. The authors show that the European hydroclimate is profoundly affected by the different scenarios, but choose to visualize the location that is among one of the least impacted. Would it not make sense to show the local differences somewhere where the signal is stronger? This modeling choice should be more clearly justified.

Equations are not correctly punctuated throughout the manuscript.

l. 7: 'depends' $\rightarrow$ 'depend'

Table 1: Should be $m^3 s^{-1}$ not $m s^{-1}$; 'statistical equilibrium' should be defined somewhere.

l. 18: What is meant by a 'major' climate tipping point? The Armstrong McKay paper uses the term 'global tipping point', and this is rigorously defined. Are you using a different term for a reason, and if so it should be defined.

l. 65: 'indirectly' and 'directly' have opposite meanings; which is it?

l. 88: 'logarthemic' $\rightarrow$ 'logarithmic'

l. 130: 'the PPD' $\rightarrow$ 'the negative PPD'

l. 144: 'eigh' $\rightarrow$ 'eight'

l. 165: 'decreases which' $\rightarrow$ 'deacreases, which'

---

## Author Comment (AC1)

**MS-No.:** egusphere-2025-1440

**Version:** Revision

**Title:** Changing European Hydroclimate under a Collapsed AMOC in the Community Earth System Model

**Author(s):** René M. van Westen, Karin van der Wiel, Swinda K.J. Falkena and Frank Selten

**Point-by-point reply to reviewer**

July 28, 2025

We thank the reviewer for their careful reading and for the useful comments on the manuscript.

*In the manuscript Changing European Hydroclimate under a Collapsed AMOC in the Community Earth System Model, eight model simulations are used to assess the impact of the AMOC strength and two different Representative Concentration Pathways (RCP4.5 and RCP8.5) on the European hydroclimate. The authors reconstruct daily potential evapotranspiration rates (PET) from CESM variables stored at monthly frequency. This allows them to diagnose the differences in dry season length and intensity under the different scenarios, as well as determining the dominant terms in the PET equation that are most sensitive to the changing climatic conditions in the eight simulations. The authors find that compared to the pre-industrial conditions (PI), nearly all scenarios show a general reduction in the potential precipitation deficit (PPD) over Europe, resulting in generally drier climate and an increase in drought extremes.*

*The eight scenarios allow for a comparison between the effects of the AMOC shutdown and the anthropogenic climate change on the European hydroclimate. Under PI conditions with reduced AMOC strength, the precipitation and PET rates decrease, and the latter dominates, resulting in dry conditions. In the RCP8.5 scenarios, the precipitation is reduced as well, but the PET rates increase, and the two effects combine, resulting in more intense and longer dry periods than in the AMOC off scenarios. Lastly, the RCP4.5 simulations provide the most important comparison between the European climate states under increased radiative fluxes with and without*

*collapsed AMOC. These results show how AMOC collapse intensifies the dry conditions in Europe in the warming climate. The RCP4.5 scenario with collapsed AMOC shows significantly longer and more intense dry season in Southern and Central Europe compared to the AMOC on case, as well as significantly increased probability for drought.*

*The study presents a meaningful and important contribution to understanding the risks associated with the combined effect of climate change and AMOC tipping on the hydroclimate. As such, the article is a relevant addition to the Hydrology and Earth System Sciences (HESS) Journal, both in terms of scientific content and result significance.*

General Comments

*The manuscript is well organized and the analysis is conducted to a high standard. Analyzing the PPD and its decomposition into precipitation and PET rates reveals the pathways driving changes in European hydroclimate across scenarios. The changes in precipitation are linked to mean sea-level pressures, and it is shown that the dominant factor affecting the PET is temperature. It is furthermore demonstrated that reconstructing $\mathrm{PET}^{\mathrm{day}}$ is important for understanding the scenario impact, as the E-P signal is weaker than P-PET , since evaporation in the model is limited by the precipitation rate.*

1. *The reconstruction of $\mathrm{PET}^{\mathrm{day}}$ and $\mathrm{PET}^{\mathrm{month}}$ in Appendix A is well justified and described. However, comparing the RMSE of end-year PPD computed from true and reconstructed PET would be a more meaningful measure of the biases introduced with the upscaling method. Since the CESM PPD biases are significant compared to ERA5, it would be useful to know to which extent the PET reconstruction may be contributing to these biases, if at all.*

   **Author's reply:**

   Following the reviewer's suggestion, we have determined the seasonally-integrated PET over the growing season as:

   $$\mathrm{PET}^{\mathrm{end}} = \int_{1\ \mathrm{Apr}}^{30\ \mathrm{Sep}} \mathrm{PET}(t')\mathrm{d}t',$$

and compared this to the seasonally-integrated $PET^{day}$ and $PET^{month}$. We also determined the yearly-integrated PET, but the results are most interesting over the growing season as PET rates are then the largest. The root-mean-square error in $PET^{end}$ is less than 30 mm over most European land surfaces and the end of growing season, boiling down to an error of a few percents as the seasonally-integrated PET is typically more than 550 mm. The introduced bias is limited and demonstrates that the upscaling method is accurate for our purposes. Note that we conduct the analysis on PET and not on PPD (as suggested by the reviewer), since the daily-averaged precipitation rates are not affected by the upscaling method and PPD biases are introduced through $PET^{day}$ or $PET^{month}$.

**Changes in manuscript:**

This additional analysis will be included and discussed in the Appendix. The seasonally-integrated PET differences will be added to Figure A1. We will also add the $PPD^{day}$ for ERA5 in Figure 2b.

2. *The communication, especially in the results section, is lacking clarity and should be improved. First of all, the significance of the k-means analysis is unclear. The authors should state clearly the significance of this analysis and how it fits with the previous results.*

   **Author's reply:**

   We agree with the reviewer that the text was difficult to follow. In particular, the simulation names were not intuitive and disrupted the flow of the narrative (as also noted in point #4 of the review). The $k$-means clustering methodology should be introduced more clearly.

   **Changes in manuscript:**

   We will revise the manuscript to improve clarity and provide a clearer introduction to the $k$-means clustering methodology in both the Methods and Results sections.

3. *More importantly, it takes a long time to realize that the focus for the reader should be the difference between the two RCP4.5 scenarios, where the role of the AMOC collapse in contributing to the hydroclimate changes under increased radiative forcing can be identified. This*

*important result drowns in the overly detailed text of the section, and is also not clearly visually indicated on the figures. The manuscript clarity would improve if both the results and the figures were streamlined to better express the focus on the differences between the results from the two RCP4.5 scenarios.*

**Author's reply:**

The reviewer is correct here, the most interesting comparison is made between the two RCP4.5 scenarios. This should be more emphasised throughout the manuscript.

**Changes in manuscript:**

We will rewrite the main text accordingly and put more emphasis on the two RCP4.5 results. We will add a statement in the Methods (section 2.1) when the CESM simulations are introduced.

4. *The naming convention for the different scenarios greatly contributes to the lack of clarity and transparency in the text and on the figures. Instead of the cumbersome 'AMOC on (FH = 0.18)', the authors could consider simply writing 'PI LOW' or 'ON LOW' to indicate the lower rate of freshwater hosing. It is not crucial for the reader to know the exact $F_H$ value throughout the text. It is also not obvious how the simulations in the manuscript are branched off from the simulations with $F_H = 0.66$ Sv (ll. 65-69). This can be clarified. Additionally, the figure titles and labels are very small, and increasing the font size would improve the ease of understanding.*

**Author's reply:**

The naming convention was indeed hard to follow and we agree with the reviewer that the exact $F_H$ value is not needed for the reader. A different naming convention should streamline the main text and we propose to label the simulations as follows: e.g., $PI_{18}^{on}$. The simulation name refers to the radiative forcing conditions, where the subscript indicates the $F_H$ strength (in units of $\times 10^{-2}$ Sv) and the superscript whether the AMOC is in its strong northward overturning state (i.e. 'on') or in its collapsed state (i.e., 'off').

**Changes in manuscript:**

We will incorporate the different naming convention in the revision, the relevant text and figures will be changed accordingly. We will also display the figures, figure captions and labels at a greater size where possible.

5. *Lastly, the discussion about model biases is lacking. The assumption that CESM biases with respect to ERA5 stay constant is highly unlikely to hold (ll. 111-113). It would be great if the authors could include more discussion about how the significant CESM biases may affect the results.*

   **Author's reply:**

   We agree with the reviewer that this assumption is not very likely, given that climate models are tuned with an AMOC on background state. We do expect, however, that the AMOC-induced changes are (much) larger than variations in climate model biases, but this cannot be tested.

   **Changes in manuscript:**

   In the revised discussion, we will expand the discussion on the different CESM biases.

Minor Comments

1. *It is surprising that the Netherlands is chosen as the reference location to show the local changes in PPD. The authors show that the European hydroclimate is profoundly affected by the different scenarios, but choose to visualize the location that is among one of the least impacted. Would it not make sense to show the local differences somewhere where the signal is stronger? This modeling choice should be more clearly justified.*

   **Author's reply:**

   We considered this location as it can be compared with the measurement station 'De Bilt'. This location is also of interest as it is situated

in Northwestern Europe, a region that shows relatively large temperature responses under a collapsed AMOC. Two other locations (Sweden and Spain) were presented in Figure A3.

**Changes in manuscript:**

We will revise this and provide a clearer justification for the location used.

2. *Equations are not correctly punctuated throughout the manuscript.*

   **Author's reply:**

   Agreed.

   **Changes in manuscript:**

   Will be corrected.

3. *l. 7: 'depends' →'depend'*

   **Author's reply:**

   Agreed.

   **Changes in manuscript:**

   Will be corrected.

4. *Table 1: Should be $m^3 s^{-1}$ not $m\ s^{-1}$; 'statistical equilibrium' should be defined somewhere.*

   **Author's reply:**

   Agreed. A statistical equilibrium is characterised by a stable climate (with time-invariant statistics) and any remaining model drift is much smaller than the internal climate variability.

   **Changes in manuscript:**

   Will be corrected and we add a definition of the statistical equilibrium in the Methods.

5. *l. 18: What is meant by a 'major' climate tipping point? The Armstrong McKay paper uses the term 'global tipping point', and this is rigorously defined. Are you using a different term for a reason, and if so it should be defined.*

   **Author's reply:**

   The reviewer is correct here.

   **Changes in manuscript:**

   This sentence will be rewritten in the revision.

6. *l. 65: 'indirectly' and 'directly' have opposite meanings; which is it?*

   **Author's reply:**

   The description of the different climate model simulations was confusing. We will also use a different naming convention to streamline the paper and update the Methods.

   **Changes in manuscript:**

   We will rewrite these sentences in the revision.

7. *l. 88: 'logarthemic' $\rightarrow$ 'logarithmic'*

   **Author's reply:**

   Agreed.

   **Changes in manuscript:**

   Will be corrected.

8. *l. 130: 'the PPD' $\rightarrow$ 'the negative PPD'*

   **Author's reply:**

   Agreed.

   **Changes in manuscript:**

   Will be incorporated as suggested.

9. *l. 144: 'eigh' → 'eight'*

   **Author's reply:**

   Agreed.

   **Changes in manuscript:**

   Will be corrected.

10. *l. 165: 'decreases which' → 'decreases, which'*

    **Author's reply:**

    Agreed.

    **Changes in manuscript:**

    Will be corrected.

---

## Author Comment (AC2)

**MS-No.:** egusphere-2025-1440

**Version:** Revision

**Title:** Changing European Hydroclimate under a Collapsed AMOC in the Community Earth System Model

**Author(s):** René M. van Westen, Karin van der Wiel, Swinda K.J. Falkena and Frank Selten

**Point-by-point reply to reviewer**

July 28, 2025

We thank the reviewer for their careful reading and for the useful comments on the manuscript.

*This study investigates the hydroclimatic response over Europe under eight climate scenarios featuring different AMOC (Atlantic Meridional Overturning Circulation) strengths, including AMOC collapse cases, across pre-industrial (PI), RCP4.5, and RCP8.5 conditions. The authors use the CESM model, which has sufficiently high resolution for this kind of regional hydroclimatic analysis. They focus on daily water balance analysis by examining daily averaged precipitation and potential evapotranspiration (PET), from which they derive the potential precipitation deficit (PPD). The results indicate that while precipitation dominates the PPD, it does not fully explain the spatial variations, and PET also plays a role—particularly as PET rates increase with warming. The study shows that AMOC collapse leads to drier conditions and increased drought extremes across all climate scenarios, with the drying effects becoming more severe under RCP4.5 and RCP8.5. Particularly noteworthy is the comparison between RCP4.5 scenarios with and without AMOC collapse, which illustrates how radiative forcing combined with an AMOC collapse significantly intensifies drying by extending the dry season.*

*Overall, this is a compelling and timely contribution, especially given the increasing interest in AMOC-related tipping points. The authors have done an excellent job in designing and executing a set of carefully structured experiments. The clarity of figures, particularly those showing the combined*

*impacts of AMOC and warming scenarios, is commendable. With minor revisions, I believe this study is well-suited for publication in HESS.*

General Comments

1. *It is intriguing that an AMOC collapse occurs under PI radiative forcing with a freshwater flux of 0.18 Sv, but not under RCP4.5 with the same flux. Further elaboration on this outcome would enhance the reader's understanding, as it suggests interesting non-linear behaviour and sensitivity to background climate states.*

   **Author's reply:**

   The description on the different simulations was confusing in the manuscript. The PI simulations were obtained from the AMOC hysteresis experiment. In this experiment, the AMOC was forced under the slowly-increasing $F_H$ at a rate of $3 \times 10^{-4}$ Sv yr$^{-1}$ (up to $F_H = 0.66$ Sv) and the AMOC collapses around $F_H = 0.525$ Sv. The $F_H$ was then reduced back to zero at the same rate and the AMOC recovers around $F_H = 0.09$ Sv. This results in a broad multi-stable AMOC regime for 0.09 Sv $< F_H <$ 0.525 Sv. Within this multi-stable regime, four simulations were branched off under constant $F_H$ and constant PI radiative forcing conditions, which are the two 'AMOC on' states and two 'AMOC off states'.

   The climate change simulations were initiated from the two AMOC on states. The RCP4.5 under the 0.18 Sv hosing did not cross the basin boundary of attraction and hence the AMOC did not collapse; this simulation was discussed in greater detail in van Westen et al. (2024, https://arxiv.org/abs/2407.19909). We do expect that a collapsed AMOC state exists under RCP4.5 and 0.18 Sv hosing, which will look similar to the collapsed AMOC state under RCP4.5 and 0.45 Sv hosing. This discussion is beyond the scope of this manuscript and is being addressed in van Westen et al. (2025, https://doi.org/10.5194/egusphere-2025-14).

   **Changes in manuscript:**

   We will rewrite section 2.1 in the revision and clarify how the different simulations were obtained.

2. *The distinction between AMOC "ON" and "OFF" states needs clarification. Figures and text indicate that both states include freshwater fluxes of 0.18 Sv and 0.45 Sv, raising questions about how the AMOC is still "ON" under such forcing. More detail on how the AMOC states are defined would be useful.*

**Author's reply:**

The description was confusing here (see also point #1 above). We will adopt a different naming convention to streamline the manuscript, e.g., $\text{PI}^{\text{on}}_{18}$. The simulation name includes the radiative forcing conditions, where the subscript indicates the $F_H$ strength (in units of $\times 10^{-2}$ Sv) and the superscript whether the AMOC is in its strong northward overturning state (i.e. 'on') or in its collapsed state (i.e., 'off').

**Changes in manuscript:**

We will rewrite section 2.1 in the revision. All the relevant text and figures will be changed accordingly.

3. *Much of Section 3.3 reads more like methodological description and could be moved to the Methods/Data section. Additionally, justification for the use of the k-means clustering approach should be strengthened. How do the results differ from a traditional analysis of MSLP changes, and why is clustering more appropriate or insightful in this case?*

**Author's reply:**

Yes, agreed. The $k$-means clustering methodology should be introduced more clearly. Part of the $k$-means clustering can be moved to the Methods.

**Changes in manuscript:**

We will incorporate these changes and provide a better justification for the $k$-means clustering in the revision.

4. *The Discussion section includes very little comparison with other studies that examine hydroclimate or European climate changes in response to AMOC shutdown. It would also be helpful to discuss how the results might change if CESM model biases were accounted for or removed.*

**Author's reply:**

Agreed, we will expand the discussion on hydroclimate responses to an AMOC shutdown. However, we note that only a few studies have provided an in-depth analysis of European hydroclimate responses

**Changes in manuscript:**

We will incorporate more literature in the revision. We will also expand the discussion on the CESM model biases.

Specific Comments

1. *Line 17–18: Replace vague or journalistic terms like "hot topic" or "major tipping point" with more scientific language.*

   **Author's reply:**

   Agreed.

   **Changes in manuscript:**

   Will be rewritten in the revision.

2. *Line 30: Specify how the seasonal cycle shifts.*

   **Author's reply:**

   There is a delayed response in the season cycle over the northern part of the Amazon Rainforest.

   **Changes in manuscript:**

   This will be clarified in the revision.

3. *Lines 26–32: It might be worth adding Saini et al., 2025 (https://agupubs.onlinelibrary.wiley.com/doi/full/10.1029/2024PA004967) here, as they examine the impact of AMOC shutdown on Australian hydroclimate. They also use a comparable higher-resolution model, and this could be another interesting region where AMOC impacts are transmitted via planetary-scale dynamics.*

**Author's reply:**

Thank you for bringing this study to our attention, this is indeed a relevant study.

**Changes in manuscript:**

The study will be incorporated in the revision.

4. *Line 62: Should read "details on how. . . "*

   **Author's reply:**

   Agreed.

   **Changes in manuscript:**

   Will be corrected.

5. *Line 195: Should read "a factor of 3.5."*

   **Author's reply:**

   Agreed.

   **Changes in manuscript:**

   Will be corrected.

6. *Lines 235–237: Grammar needs correction here. Consider rephrasing for clarity and correct sentence structure.*

   **Author's reply:**

   Agreed.

   **Changes in manuscript:**

   Will be rewritten in the revision.

7. *Lines 303–305: Please indicate which scenario (e.g., RCP8.5) is being discussed to ensure clarity.*

   **Author's reply:**

   Agreed.

**Changes in manuscript:**

The two RCP8.5 scenarios will be mentioned here.

8. *Lines 332–333: This sentence appears to repeat the introductory paragraph of the section. Consider removing or rephrasing to avoid redundancy.*

   **Author's reply:**

   Agreed.

   **Changes in manuscript:**

   Sentence will be removed in the revision.

---

## Author Response (AR2)

**MS-No.:** egusphere-2025-1440

**Version:** Revision II

**Title:** Changing European Hydroclimate under a Collapsed AMOC in the Community Earth System Model

**Author(s):** René M. van Westen, Karin van der Wiel, Swinda K.J. Falkena and Frank Selten

**Point-by-point reply to the Editor**

*I thank the authors for their thorough revisions. Following the re-assessment by Referee 2 and my own evaluation, the paper can be accepted for publication subject to the following minor changes.*

We thank the Editor for the final comments on the manuscript.

1. *Please separate the Section "Summary and Discussion" into two distinct sections: first "Discussion" (interpretive parts) and then "Summary" (broader consequences and conclusions), following the standard structure of scientific papers. This should be feasible without fundamental changes to the exiting material but rather some re-arranging and transitional edits – for example, by using the first and last paragraphs of current Section "Summary and Discussion" as a starting point for the new "Summary" section, while ensuring the text flows well over both of the new sections and fits their respective purposes.*

   **Author's reply:**

   Agreed, this section can be split into two parts, following the standard structure of scientific papers.

   **Changes in manuscript:**

   The revised manuscript now includes a separate Discussion Section and Summary Section. Textual changes were kept to a minimum.

2. *I appreciate the authors' effort to improve readability of the figure labels; however, the y axis labels and the y axis titles are still very small and difficult to read in most figures. I understand there is a balance to strike between actual figure content and labels, but please improve these*

*as well. For reference, Figure 2 is adequately readable, and the font sizes should not be substantially smaller for good readability.*

**Author's reply:**

We have further improved all figures for readability. To be specific, labels and ticks for the axes, and legend labels are displayed at a larger size.

**Changes in manuscript:**

All figures are revised accordingly.

3. *Figure A1: Please move the equations from the figure caption into the appendix text, and in the caption only mention the abbreviations "RMSE" / "RMSEend".*

**Author's reply:**

Agreed.

**Changes in manuscript:**

The equations are moved to the appendix text, abbreviations are mentioned in the caption.